# Sorption of Halogenated Anti-Inflammatory Pharmaceuticals from Polluted Aqueous Streams on Activated Carbon: Lifetime Extension of Sorbent Caused by Benzalkonium Chloride Action

**Barbora Kamenická, Tomáš Weidlich * and Miloslav Pouzar**

Chemical Technology Group, Institute of Environmental and Chemical Engineering, Faculty of Chemical Technology, University of Pardubice, Studentska 573, 532 10 Pardubice, Czech Republic; barbora.kamenicka@upce.cz (B.K.); miloslav.pouzar@upce.cz (M.P.)
* Correspondence: tomas.weidlich@upce.cz; Tel.: +420-46-603-8049

**Abstract:** The enhancement of the adsorption capacity of activated carbon (AC) using benzalkonium chloride (BAC) within the adsorption of halogenated pharmaceuticals flufenamic acid (*flufa*) and diclofenac (*dcf*) was investigated in this study. An adsorption kinetic study was performed to evaluate the adsorption mechanisms. The adsorption mechanism of both drugs on granulated AC as well as saturated AC activated by BAC can be evaluated via pseudo-second kinetic order. The equilibrium adsorption capacity of spent granulated AC in co-action with BAC ($q_{flufa}$ = 195.5 mg g$^{-1}$ and $q_{dcf}$ = 199.5 mg g$^{-1}$) reached the adsorption capacity of virgin granulated AC ($q_{flufa}$ = 203.9 mg g$^{-1}$ and $q_{dcf}$ = 200.7 mg g$^{-1}$). Finally, batch and column arrangements were compared in an effort to possible practical application of exhausted AC in co-action with BAC. In both column and batch experiments, adsorption capacities of spent granulated AC for *flufa* increased using BAC by 170.4 mg g$^{-1}$ and 560.4 mg g$^{-1}$, respectively. The proposed mechanism of adsorption enhancement is the formation of less polar ion pairs and its better affinity to the non-polar AC surface. The drug concentrations were determined using the voltammetric method on carbon paste electrodes. The formation of ion pairs has been studied by the H$^1$ NMR technique, and solubility in water of drugs and respective ion pairs were investigated using octan-1-ol/water coefficients ($P_{OW}$).

**Keywords:** adsorption; activated carbon; flufenamic acid; diclofenac; halogenated pharmaceutical; carboxylic acid; benzalkonium chloride; quaternary ammonium salt; ionic liquid





## 1. Introduction

Pharmaceuticals have been classified as one of the most significant groups of environmental pollutants [1]. A particular group of pharmaceuticals is the family of non-steroidal anti-inflammatory drugs (NSAIDs), such as diclofenac (*dcf*) and flufenamic acid (*flufa*); see Figure 1 below. *dcf* is a widely used NSAID in many countries around the world as a commercial medicament (e.g., Voltaren or Olfen) [2]; *dcf* is structurally and functionally similar to *flufa*. This non-prescription drug is also sold in Japan, China, India, Spain, or Germany as commercial preparations (e.g., Flufen or Opyrin) [3]. Consequently, NSAIDs are among the most detected drugs in the aquatic environment [1]. In particular, the manufacture of NSAIDs generated a large amount of wastewater with a high content of produced medicaments [4].

**Figure 1.** The structure of sodium salt of diclofenac (*dcf*) and flufenamic acid (*flufa*).

Due to the fact that biological treatment in wastewater treatment plants (WWTPs) is often not efficient [5], several methods have been applied for the degradation of halogenated pollutants, such as oxidation, photocatalytic degradation, or membrane processes [1]. There are also several innovative methods of hydrodechlorination of halogenated compounds using aluminum-based alloys [6,7].

However, activated carbon (AC) has been extensively used as an effective adsorbent in the separation of a wide variety of organic contaminants from wastewater [8]. The efficient adsorption of pharmaceuticals, dyes, or pesticides on ACs is well documented in many publications [9]. The main reasons for the mentioned applications are the high surface area (800–1500 $m^2$ $g^{-1}$), well-developed internal microporosity, and a wide spectrum of functional groups bound at the surface [8]. AC can be utilized as a powdered or granulated form in both batch and fixed-bed column technological arrangements [10].

After exhausting the adsorption capacity of AC, disposal of spent AC is required [10]. The disposal of spent AC depends on the type of adsorbed pollutants (volatile vs. non-volatile), technical applicability or operating conditions, and economic costs [11]. There are several different methods that can be considered as the processes for the proper disposal of spent AC; incineration in a hazardous waste incinerator or regeneration (or reactivation) [11].

The total annual world consumption of AC is ca. 5.4 million metric tons, and the annual consumption of AC in west Europe is approximately 2 million metric tons [12]. In addition, the company-specific average quantity is about 110 tons per company per year [11]. Approximately more than 50% of the total spent AC is incinerated instead of reused, although regeneration (or reuse) is always a desirable option in view of environmental protection and economic perspective [11]. In technology practice, thermal (pyrolytic) desorption is the most applied for AC regeneration used to remove non-volatile contaminants [11].

The cost of virgin AC ranges from \$340 to \$1200 per ton [13], and the cost of purchasing regenerated carbon ranges from \$1100 to \$1720 per ton [14]. Furthermore, many countries in Europe do not anticipate regeneration (desorption) facilities; thus, the transport of saturated sorbents (dangerous waste) and operating costs increase. One option for decreasing the price of AC regeneration is to modify these adsorbents, which can lead to increased AC adsorption capacity [15]. The lifetime of AC can be extended via modifications. In addition, the modification of AC can also tailor specific physical and chemical properties and improve its affinity for polar organic pollutants in water [8].

As presented in the available literature [15–23], chemically modified activated carbon showed the maximum adsorption capacity toward different types of organic pollutants from an aqueous solution. While many authors have reported the acidic, basic, or other treatments (oxidation, modification with metallic cations such as Ag, Fe, etc.) of AC as a possible way to improve the adsorption of organic pollutants from water [8,15–17], only several authors published the impregnation/modification methods of AC using quaternary ammonium salts ($R_4NX$; also surfactants or ionic liquids) [18–23]. A consensus among the published studies is that hydrophobic interactions between surfactants and AC were the primary mechanism that govern the adsorption of surfactants on AC [17]. Kuang et al.

reported the enhanced adsorption of cationic dye on AC modified with anionic surfactant since the main mechanism of adsorption denotes ionic effects [24]. Parette and Cannon [23] presented a recent approach to modification of AC—granulated AC was pre-modified with cationic surfactants to improve perchlorate removal from water in column experiments.

Our research group previously [25] proved that the modification of carbonaceous adsorbents using $R_4NX$ can increase the efficiency of the removal of organic acids and their salts from the aqueous solution. The $R_4NX$s are ionic compounds that include large organic cations and inorganic anions, where R is an alkyl chain, and X is an anion, typically halide $Cl^-$, $Br^-$ [26]. The method is based on ion exchange between added $R_4NX$ and organic acidic pollutants, resulting in the formation of respective ion pairs [27]. Benzalkonium chloride (BAC; its chemical structure is depicted in Scheme 1) is also representative of quaternary ammonium compounds. As described in the literature [28], there are many applications of BAC as disinfectants, cleaning products or other surfactants, preservatives, hand lotions, cosmetics, biocides, or phase-transfer catalysts in organic chemistry. BAC is mostly available as 0.01–5% aqueous solution. However, it may be provided as 25–50% concentrated aqueous solutions [29].

**Scheme 1.** The formation of the ion pair BAC-*dcf*.

In the present work, spent activated carbon was modified with commercially available BAC to enhance the adsorption capacities of saturated AC. A simple modification method is proposed to prolong the lifetime of AC and decrease the economic costs of the adsorption of the halogenated anti-inflammatory drugs from wastewater.

## 2. Materials and Methods

### 2.1. Chemicals

To prepare 2 mmol $L^{-1}$ stock solutions of tested pharmaceuticals, the appropriate quantity of *dcf* (sodium salt; Sigma-Aldrich, St. Louis, MO, USA; purity higher than 98%) or *flufa* (Sigma-Aldrich, USA; purity higher than 98%) was dissolved in aqueous NaOH solution (400 mg $L^{-1}$). The 50% aqueous solution of BAC was purchased from Sigma-Aldrich, USA (density approx. 0.98 g $cm^{-3}$, average molar mass 283.88 g $mol^{-1}$, solubility in water 100 mg $mL^{-1}$). In individual experiments, stock solutions of *dcf*, *flufa*, and BAC were diluted as required using demineralized water. Octan-1-ol and DMSO-$d_6$ were purchased from Merck (Praha, Czech Republic) in purity higher than 98%. Additional chemicals and solvents in p.a. quality were obtained from a local supplier (Lach-Ner Co., Neratovice, Czech Republic).

### 2.2. Adsorbents

Powdered active carbon (PAC) Silcarbon CW20 was purchased from Brenntag Co. (Praha, Czech Republic). Granular active carbon (GAC) Hydraffin CC8x30 was obtained from Donau Carbon GmbH (Frankfurt am Main, Germany). The specification of both types of virgin ACs is presented in Table 1.

**Table 1.** The specifications of applied activated carbons according to suppliers [30,31].

| Characteristics | PAC | GAC |
|---|---|---|
| Total surface area (BET-method)/m$^2$ g$^{-1}$ | 1300 | 1150 |
| Granulation (mesh) | <45 μm | 0.6–2.36 mm |
| Ash content/wt. % | 8 | <4 |
| Water content/wt. % | <10 | <5 |
| Iodine adsorption/mg g$^{-1}$ | >900 | >1000 |

Impregnated GAC (Hydraffin CC8x30) was prepared by vigorous stirring of 20 g of virgin GAC with 100 mL of 5% aqueous solution of BAC for 20 h, its separation using filtration, rinsing with 100 mL of water and drying to dryness. Based on the content of cationic surfactants before (52.1 g L$^{-1}$) and after (14.5 g L$^{-1}$) impregnation, 72.2% removal of BAC on GAC was observed.

The sample of exhausted GAC Hydraffin CC8x30 (spent-GAC) used in industrial column adsorber was purchased from a local chemical company. This spent-GAC was obtained by the adsorption of a mixture of acidic organic contaminants from industrial wastewater in a large-scale fixed-bed column arrangement. The determined iodine number value of spent-GAC was 188 mg L$^{-1}$). The leaching experiment of spent-GAC (5 g of spent-GAC + 200 mL of water) indicated the release of 5.1 mg COD$_{Cr}$ per litter of water from 1 g of spent-GAC and no signal in voltammogram (see Figure S1 in Supplementary Materials).

The samples of spent as well as virgin GAC were also characterized via ED XRF analysis. Figure S8a,b in Supplementary Materials illustrates the results of ED XRF analyses. Both samples of GAC saturated with DCF and GAC saturated with the ion pair BAC-DCF approved significantly higher content of chlorine compared with virgin GAC or spent-GAC (used in industrial column adsorber). On the other hand, spent-GAC from industrial column adsorber contains higher quantities of calcium and iron descended from treated industrial wastewater.

## 2.3. Adsorption Experiments and Ion Exchange

Adsorption batch experiments were carried out in magnetically stirred 250 mL round-bottomed flasks at 25 °C using Starfish equipment installed on an electromagnetic stirrer Heidolph HeiStandart with a temperature sensor Pt1000. The appropriate quantity of BAC and AC was added to the 250 mL of model wastewater containing respective drugs at $c_0 = 100$ mg L$^{-1}$. After an appropriate time of vigorous stirring (at 400 rpm), the suspensions were immediately filtered. For detailed settings of adsorption kinetics experiments, see Scheme S1 in Supplementary Materials.

The column experiments were carried out in a glass laboratory column (length 14 cm, diameter 4.5 cm) filled with granulated activated carbon. The column was loaded with the appropriate solution, which percolated downward. The flow rate was maintained at 8.4 cm$^3$ min$^{-1}$. The mass of GAC applied in the column was 25 g, and the bed depth of GAC was 6.5 cm.

Finally, for the prove of ion pairs production, the 2.5 mmol of respective NSAIDs was stirred with 5 mmol of BAC in 500 mL H$_2$O (2-times excess of R$_4$NX per 1 mol of -COO$^-$ group bound in NSAIDs). Then, the reaction mixture after ion pair formation was extracted with one portion of CH$_2$Cl$_2$ (100 mL) overnight under vigorous stirring and subsequently evaporated from the obtained dichloromethane phase to dryness.

## 2.4. Evaluation of Adsorption

The removal efficiency (%) of NSAIDs from the model aqueous solutions was calculated using relationship (1):

$$\eta = \left(1 - \frac{c}{c_0}\right) \times 100 \tag{1}$$

where $\eta$ is the removal efficiency of NSAIDs (%), $c$ is the concentration of NSAIDs after adsorption (mmol L$^{-1}$), and $c_0$ is the initial concentration of NSAIDs (mmol L$^{-1}$). All the error bars in the figures are calculated as the relative standard deviation (RSD) of decoloration efficiency. In all cases, the RSD was less than 6%. Differences in adsorption processes were evaluated according to the equilibrium adsorption capacity described in Equation (2) [32].

$$q_e = \frac{(c_0 - c_e) \times V}{m} \tag{2}$$

where $c_0$ is the initial concentration of NSAIDs (mg L$^{-1}$), $c_e$ is the equilibrium concentration (mg L$^{-1}$), $V$ is the volume of drug aqueous solution (L), and $m$ is the dosage of adsorbent (g).

The pathway of adsorption and its kinetic parameters can provide fundamental information with respect to the mechanism of the adsorption process [32–34]. To investigate the adsorption, two most applied conventionally kinetic orders are being tested to fit the experimental data: pseudo-first-order kinetic model (PFO) and pseudo-second-order kinetic model (PSO), respectively [33]. In addition, the interparticle diffusion model was also evaluated [35,36]. The equations of the above-mentioned kinetic adsorption models are presented in Equations (3)–(5) in Table 2.

**Table 2.** Kinetic adsorption models were applied for the evaluation of the adsorption mechanism.

| Equation No. | Kinetic Model | Equation * |
|:---:|:---:|:---:|
| (3) | PFO | $ln(q_e - q_t) = lnq_e - k_1 t$ |
| (4) | PSO | $\frac{t}{q} = \frac{1}{k_2 q_e^2} + \frac{1}{q_e} \cdot t$ |
| (5) | interparticle diffusion | $q = k_p t^{1/2} + \lambda$ |

* Note: $t$ is time (min), $q_e$ is equilibrium adsorption capacity (mg g$^{-1}$), $q_t$ is adsorption capacity at time $t$ (mg g$^{-1}$); $k_1$ (min$^{-1}$) is PFO rates constant; $k_2$ (g mg$^{-1}$ min$^{-1}$) is PSO rate constant; $k_p$ (mg g$^{-1}$ min$^{-\frac{1}{2}}$) is constant related to the diffusion coefficient and $\lambda$ is the intercept of the intraparticle diffusion model.

The adsorption in columns influences many factors, such as the characteristics of adsorbate and adsorbent, the particle size of adsorbent, concentration of adsorbate, flow rate, and bed depth [37]. In this elementary column study, we mainly focused on the possibilities of sorption bed reactivation with the aim of extending the sorbent lifetime. Plots of the ratio of $c_{effluent}/c_{influent}$ versus treated volume are denominated as a breakthrough curve [38]. The breakthrough time is defined as the time when the $c_{effluent}/c_{influent} = 0.5$, and it can be expressed by following Equation (6) [38,39].

$$t_{50} = \frac{V}{F} \tag{6}$$

where $t_{50}$ is breakthrough time (min), $V$ is treated volume (dm$^3$), and $F$ is flow rate (cm$^3$ min$^{-1}$). Breakthrough capacity $q_{column}$ at ($c_{effluent}/c_{influent} = 0.5$) can be expressed in mg of adsorbed drug per gram of adsorbent and calculated using the following Equation (7) [38].

$$q_{column} = \frac{\sum_{0.5} drug\ adsorbed\ on\ AC}{m_{AC}} = \frac{\sum_{0.5} V \times (c_i - c_e)}{m_{AC}} \tag{7}$$

where $m_{AC}$ is the mass of AC in column (g), $V$ is treated volume (L), $c_i$ is the influent concentration of drug (mg L$^{-1}$), and $c_e$ is the effluent concentration of drug (mg L$^{-1}$). For a detailed theoretical background of fixed-bed column adsorption studies, see ref. [39].

## 2.5. Determination of Octan-1-ol/Water Partition Coefficient

An aqueous solution containing 1 mmol of NSAIDs was introduced to the round-bottomed flask (in case of log $P_{OW}$ of ion pairs, 1 mmol of R$_4$NX per 1 mmol of -COO$^-$ group bound in the drug was added), and the total volume of aqueous phase was adjusted to

100 mL with water and the mixture was fulfilled using 100 mL of octan-1-ol. The prepared two-phase mixture was agitated at 400 rpm overnight, the immiscible phases were separated in a separatory funnel, and a concentration of *flufa/dcf* or respective ion pairs was analyzed using voltammetry. The partition coefficient $P_{OW}$ was calculated according to Equation (8). Each experiment was performed three times; the presented values of log $P_{OW}$ are calculated as mean values. The error bars in the figures dealing with octan-1-ol/water partition coefficients presented the standard deviation of the log $P_{OW}$ value.

$$\log P_{OW} = \frac{c_{octanol}}{c_{water}} \tag{8}$$

### 2.6. Chemical Analysis

The electrochemical measurements of *flufa* or *dcf* were carried out using an AUTO-LAB analyzer (PGSTAT-128N, Autolab/Metrohm, Oss, The Netherlands/Herisau, Switzerland) coupled with a three-electrode cell (working carbon paste electrode—CPE, reference Ag/AgCl/KCl electrode and Pt-plate auxiliary electrode). The CPE was prepared via hand-homogenization of 0.5 g graphite powder and 0.3 mL of paraffin oil. The homogeneous paste was manually filled into a Teflon electrode body with the piston. The surface of the CPE was in situ modified with cetyltrimethylammonium bromide (CTAB) and renewed by wiping off a thin layer of the carbon paste with a wet filter paper. The measurements were carried out under the conditions: square wave voltammetry (SWV); 0.1 mol $L^{-1}$ PBS + 100 µmol $L^{-1}$ CTAB; potential scan 0.2–1.2 V vs. ref.; SWV-ramp: frequency 60 Hz and amplitude 40 mV; scan rate of 300 mV $s^{-1}$. For a detailed description of the determination method, see ref. [40,41].

$^1$H NMR spectra were recorded in DMSO-$d_6$ solutions on a Bruker Avance 500 spectrometer (equipped with a Z-gradient 5mm Prodigy$^{TM}$ cryoprobe) at a frequency of 500.14 MHz or on a Bruker UltraShield$^{TM}$ 400 spectrometer at a frequency of 400.13 MHz at 295 K. The solutions were obtained by dissolving dried ion pairs (resulting from the work-up of reaction mixtures) in 1 mL of DMSO-$d_6$. The values of $^1$H chemical shifts were calibrated to the residual signals of DMSO-$d_6$ ($\delta(^1$H) = 2.50 ppm). The mutual content of constituents in obtained mixtures was established using the integration of the corresponding area of resonances.

Analyses of the $COD_{Cr}$ were carried out in accordance with the ISO EN 9562 standard, and in aqueous solutions, it was determined by the Hach Lange cuvette test using a Hach DR2800 (Austria) VIS spectrometer. The content of BAC in reaction mixtures was determined using cationic surfactants, the Hach Lange cuvette test, and the Hach DR2800 (Austria) VIS spectrometer according to the ISO 8466-1 standard.

ED XRF measurements were performed using the ED XRF spectrometer ElvaX (Elvatech, Kyiv, Ukraine). X-ray tube with Pd anode was operated at a current of 50 µA and voltage of 10 kV for the light element spectral region (Na–Ti) and 45 kV for the heavy element spectral region (V–U). Helium micro-flushing of the sample chamber was used for measurements in light element spectral region to suppress the intensity of the Ar signal and increase the sensitivity for measured elements. The acquisition time of 90 s was used for both spectral regions. Samples of coal granulate were simply poured into a sample cup covered with Mylar foil without any kind of other sample preparation step.

## 3. Results and Discussion

### 3.1. Adsorption of Halogenated Drugs on Activated Carbon

The separation of two NSAIDs, *flufa* and *dcf*, using adsorption on different types of ACs was investigated. The effect of dosage of powdered, granulated, and spent granulated ACs was compared. For these experiments, the initial concentration of medicaments $c_0$ = 100 mg $L^{-1}$ was applied. As Mascolo et al. [42] presented, the concentration of pharmaceuticals in real technological wastewater can be even greater than 300 mg $L^{-1}$. As several authors presented [43–45], adsorption can show different behavior for micropollutants in the aqueous environment [46] and concentrated aqueous streams from

the production of organic species [42]. Therefore, the mentioned initial concentration $c_0 = 100$ mg $L^{-1}$ was used for (i) simulation of concentrated aqueous solutions of *flufa* and *dcf*—represented wastewater streams from the manufacture of halogenated drugs, as well as for (ii) evaluation of adsorption mechanisms that better expressed the real conditions.

The published aqueous solubility of *flufa* and *dcf* in water is 100 mg $L^{-1}$ and 237 mg $L^{-1}$, respectively [47,48]. In addition, increasing the pH above 8 and temperature at 310 K increases the solubility of tested drugs in water [49]. Based on these facts, the stock solutions were prepared in an aqueous solution of NaOH at constant pH = 8.5.

The results of adsorption experiments are presented in Figure 2. The dosage of 0.8 g of both PAC and GAC per one litter of 100 mg $L^{-1}$ *dcf* or *flufa* aqueous solution enables the highest removal efficiency (>99.9%) after 20 h action. On the other hand, the compromise between the dosage of the above-mentioned carbonaceous adsorbents and the satisfactory removal efficiency can be provided using 0.4 g per liter of 100 mg $L^{-1}$ respective drug solution.

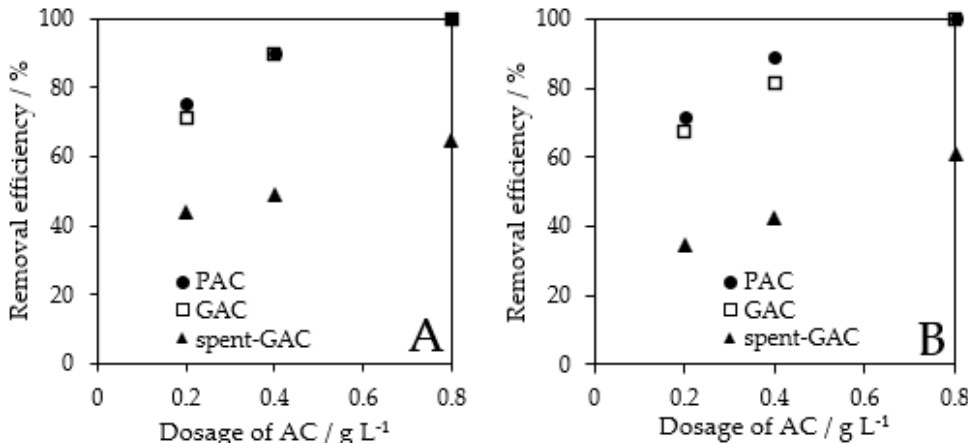

**Figure 2.** A dependence of removal efficiency of *flufa* (**A**) and *dcf* (**B**) from aq. solution (100 mg $L^{-1}$) on the quantity of used ACs (after 20 h of action).

As can be expected, the adsorption of both pharmaceuticals on spent-GAC (for description, see the Experimental Section 2.1) enables lower removal efficiencies than commercial ACs; see Figure 2. It is probably caused by the saturation of most of the specific area of spent-GAC, which leads to a lowering of adsorption capacity. For a better comparison of the adsorption processes, the optimal dosage of spent-GAC was also set at 0.4 g of spent-GAC per one litter of aqueous solution of the respective drug.

Afterward, the rates of *flufa* and *dcf* adsorption on PAC, GAC, and spent-GAC at the optimal doses were investigated. We observed that the equilibrium state for the adsorption of drugs on PAC was achieved after 90 min; see Figure 3. On the other hand, virgin GAC and a sample of spent-GAC provide the adsorption equilibrium after one hour. These differences can be caused by interaction forces, adsorption capacities, or specific surface areas of powdered and granulated ACs [11–13].

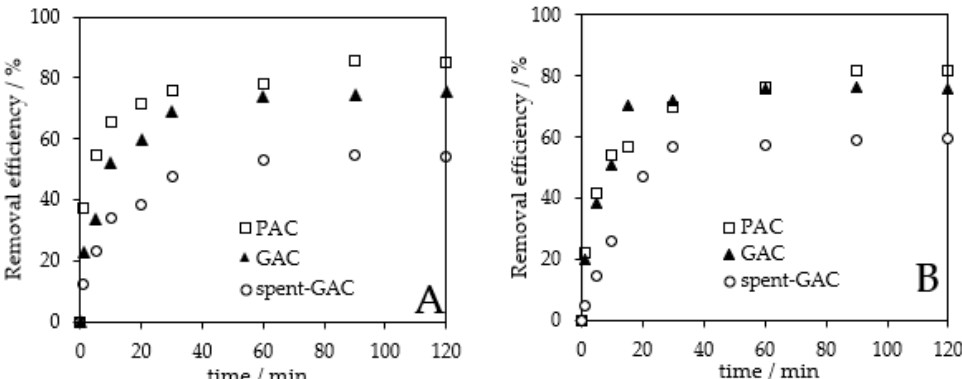

**Figure 3.** Adsorption rates of *flufa* (**A**) and *dcf* (**B**) removal from 100 mg $L^{-1}$ aq. solution using different types of ACs (0.4 g $L^{-1}$).

*3.2. Enhanced Adsorption Using BAC*

Our previous works [25,27,40,41,50] proved that the addition of $R_4NX$ to the aqueous solutions of acidic halogenated compounds (-COO⁻ od -SO₃⁻ groups), including dyes or drugs caused the ion exchange and the formation of respective insoluble ion pairs (Contaminant-COONR₄, see Equation (9)) that can be separated from aqueous solutions by simple filtration.

$$\text{Contaminant-COONa} + R_4NX \rightarrow \text{Contaminant-COONR}_4 + NaX \tag{9}$$

Therefore, the preliminary experiments tested the possibilities of separation of *dcf* and *flufa* using BAC ($R_4NX$) action alone. BAC combines both sufficient aqueous solubility and removal efficiency of acidic halogenated pollutants; for more information, see our previous study [25,50].

A separation rate of *flufa* and *dcf* from an aqueous solution (100 mg $L^{-1}$) using BAC is illustrated in Figure 4. Added 0.4 g $L^{-1}$ of BAC enables the removal of up to 52% of *flufa* and 41% of *dcf*, respectively. The higher separation efficiency of *flufa* corresponds to the higher octan-1-ol/water partition coefficient $P_{OW}$ determined for these drugs (and probably also to the lower solubility of *flufa* in water [47,49]); see Figure S2 in Supplementary Materials. Furthermore, using BAC, we also observed rapid precipitation of both *dcf* and *flufa*. The equilibrium of the separation process of both tested pharmaceuticals was achieved after 30 min; see Figure 4.

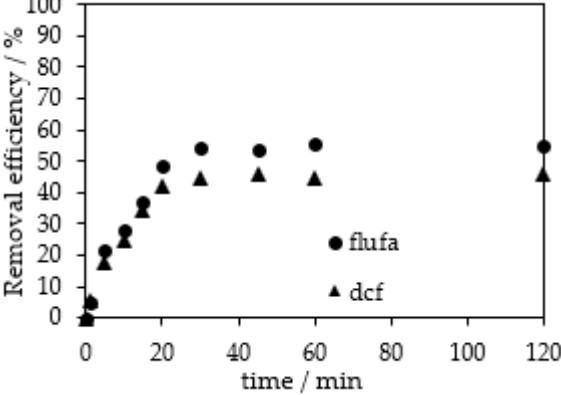

**Figure 4.** The separation rate of drugs ($c_0$ = 100 mg $L^{-1}$) using the action of BAC (0.4 g $L^{-1}$) alone.

In addition, rapid ion exchange according to the reaction in Scheme 1 was also proved by the formation of the ion pair BAC-*dcf* and its separation using dichloromethane and NMR analysis (for NMR spectra; see Figure S3 in the Supplementary Materials). Analogi-

cally, the formation of the ion pair of BAC with *flufa* was also proved; see Figure S4 in the Supplementary Materials.

Taking into account the cost of adsorption processes, it would be worth considering the possibility of enhancement of the adsorption capacities of used sorbents. Therefore, the possibilities of increasing both GAC and spent-GAC adsorption capacity using BAC were investigated in the additional step.

First, we compared the different methods (three methods A–B) for enhancing the sorption capacity of virgin GAC by the action of BAC. The methods comprise (method A) preliminary formation of the ion pair BAC-*flufa* and subsequent addition of virgin GAC to formed ion pair; (method B) adsorption of *flufa* on GAC impregnated with BAC (see specifications in Experimental Section 2.1) and (method C) in situ production of the ion pair BAC-*flufa* during adsorption on virgin GAC (co-action of BAC with GAC) were compared. For results and graphical presentation of described methods A–C, see Figure S5 in Supplementary Materials.

As can be seen in Figure S5A in Supplementary Materials, impregnated GAC (method B) allows 37% lower removal efficiency of *flufa* from aqueous solution than virgin GAC alone. We can estimate that the lower efficiency of *flufa* adsorption on impregnated GAC could be caused by the clogging of the specific surface area of GAC with BAC [25]. With respect to these results (and also previously published results [25]), this time-consuming impregnation method was not therefore further investigated.

On the other hand, rapid ion exchange and ion pair formation of BAC-*flufa* and subsequent adsorption of these ion pairs on GAC (method A) reached almost 100% removal of *flufa* from aqueous solutions; see Figure S5 in Supplementary Materials. These great removal efficiencies can be explained by the better affinity of less polar ion pairs to the non-polar surface of GAC. In this study, the lower polarity of ion pairs BAC-*dcf* and BAC-*flufa* than the origin sodium salts of *dcf* and *flufa* is proved according to the octan-1-ol/water coefficients (BAC-*flufa* $P_{OW}$ = 5.13 and BAC-*dcf* $P_{OW}$ = 1.73); see Figure S2 in the Supplementary Materials. In addition, BAC alone can be well adsorbed on virgin GAC. We observed the 72.2% removal of cationic surfactants from a used aqueous solution of BAC within the GAC impregnation; for more details, see Experimental Section 2.1. It is also in good agreement with Polamor et al. [51].

Finally, the co-action of BAC with GAC in aqueous solutions of *flufa* was tested (method B). This method enables almost similar results, such as the preliminary formation of ion pairs and its subsequent adsorption on GAC (method A). In addition, the direct co-action of BAC and GAC (method C) is less time-consuming and technologically undemanding. The high adsorption capacities of GAC and BAC co-action (method C) can be explained by the parallel effect of (i) conventional adsorption of pharmaceuticals on the GAC surface, (ii) the ion exchange reaction between the BAC and acidic drugs, and again (iii) the high affinity of the in situ formed ion pairs to the GAC surface.

According to the cationic surfactant tests, the content of BAC after in situ modification of GAC and adsorption of *flufa* is not above the allowed limits of surfactants in sewage waters [52], and BAC does not remain in the aqueous solution. It also established the above-described in situ formation of ion pairs BAC-*flufa*. Therefore, these treated aqueous streams can be discharged into the biological WWTP.

Instead of virgin GAC, the spent-GAC can be modified in the same way; see Figure S5 in Supplementary Materials. The removal of *flufa* from aqueous solution using co-action of spent-GAC with BAC (81.3%) almost reaches the removal efficiency on virgin GAC alone (89.6%) after 20 h of action. Moreover, a co-action of spent-GAC with BAC increased the removal efficiencies of *flufa* up to 25% compared to spent-GAC alone.

### 3.3. Adsorption Kinetics Study

The evaluation of adsorption kinetics can provide important information with respect to the adsorption process [35]. In addition, predicting the rate of adsorption is probably the most important factor in adsorption system design [35]. To investigate the adsorption,

three kinetic models were tested to fit the experimental data PFO, PSO, and interparticle diffusion adsorption kinetics models. The data were modeled according to refs. [32–35,53].

The earlier assumptions described that (i) the PFO kinetic model explains rather the physical adsorption mechanism when the rate-limiting step of adsorption is the spontaneous diffusion of selected pollutant to the surface of the adsorbent [32] and (ii) the PSO kinetic model describes an adsorption pathway based on chemical bonds between pollutant and functional groups present at the adsorbent surface [33]. However, several researchers [54] claimed that adsorption mechanisms cannot be directly assigned based on observing simple kinetic experiments or by fitting kinetic models (i.e., the PFO and PSO models). Adsorption mechanisms can be potentially established by several analytical methods, i.e., FTIR [54].

As we described in the previous chapter, the best results of impregnation of GAC using BAC provided the direct co-action of BAC and GAC or spent-GAC (method C). Other impregnation methods (methods A and B) of GAC using BAC were not further investigated. Therefore, within the adsorption kinetics study, we compared the adsorption of both tested drugs on GAC and spent-GAC alone and the co-action of BAC with GAC or spent-GAC. The research scope of these experiments is summarized in Scheme S1 in Supplementary Materials.

The PFO and PSO parameters were evaluated from the experimental data, which are shown as the dependence of the adsorption capacity on time; see Figure 5. The calculated parameters of the kinetic models of PFO and PSO are listed in Table 2. It was observed that the PSO kinetics adsorption model fits well with linear function and describes the *dcf* and *flufa* adsorption more correctly, especially for carbonaceous adsorbents alone. It is in good agreement with Azazian et al. [34] that most of the adsorption kinetics data can be evaluated according to the PSO model. The fitting of experimental data with the PFO and PSO model within the adsorption of *flufa* is also presented in Figure S6 in Supplementary Materials.

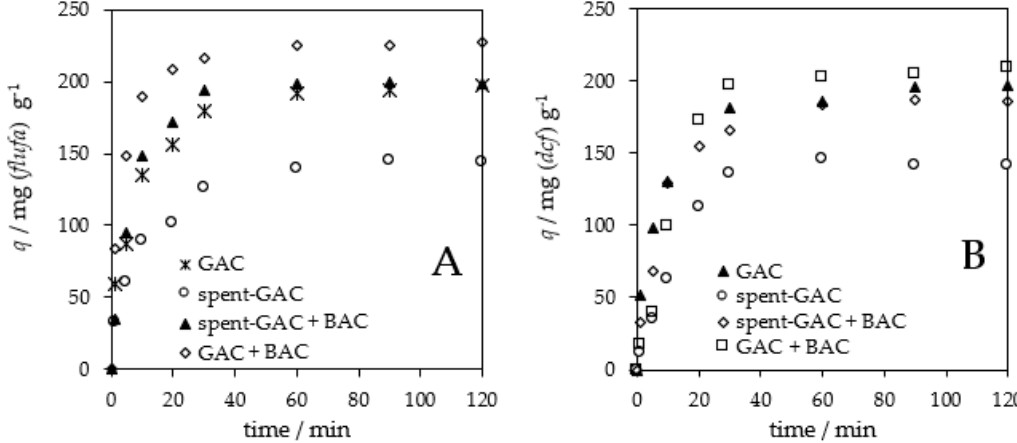

**Figure 5.** Adsorption rate of *flufa* (**A**) and *dcf* (**B**) removal from aq. solution ($c_0$ = 100 mg L$^{-1}$) on different types of GAC (0.4 g L$^{-1}$) and in co-action of BAC (0.2 g L$^{-1}$) and GAC (0.4 g L$^{-1}$).

The calculated parameter $q_e$ of the PSO model is almost similar to the experimental data (for comparison, see Figure 5 and Table 3). Clearly, the equilibrium adsorption capacities decrease in the following order: GAC in co-action of BAC > GAC > spent-GAC in co-action of BAC > spent-GAC. Virgin GAC provides the equilibrium adsorption capacity of around 200 mg g$^{-1}$ for both *flufa* and *dcf*. On the other hand, spent-GAC enables the equilibrium adsorption capacity to be only approx. 160 mg g$^{-1}$. However, the $q_e$ for in situ modified spent-GAC with BAC reaches a value comparable with that of virgin GAC.

**Table 3.** Parameters of evaluated adsorption kinetic models.

| Kinetic Model | Kinetics Constants | flufa | | | | dcf | | | |
|---|---|---|---|---|---|---|---|---|---|
| | | GAC | Spent-GAC | Spent-GAC + BAC | GAC + BAC | GAC | Spent-GAC | Spent-GAC + BAC | GAC + BAC |
| PFO | $k_1 / \min^{-1}$ | 0.062 | 0.062 | 0.086 | 0.172 | 0.076 | 0.070 | 0.075 | 0.098 |
| | $q_e / \text{mg g}^{-1}$ | 166.8 | 123.1 | 168.3 | 193.9 | 129.6 | 151.5 | 159.7 | 217.7 |
| | $R^2$ | 0.9851 | 0.9616 | 0.9120 | 0.9747 | 0.9668 | 0.9880 | 0.9617 | 0.9913 |
| PSO | $k_2 / \text{g mg}^{-1} \min^{-1}$ | $1.0 \times 10^{-3}$ | $0.9 \times 10^{-3}$ | $1.2 \times 10^{-3}$ | $3.1 \times 10^{-3}$ | $1.4 \times 10^{-3}$ | $0.4 \times 10^{-3}$ | $0.9 \times 10^{-3}$ | $0.4 \times 10^{-3}$ |
| | $q_e / \text{mg g}^{-1}$ | 203.9 | 153.6 | 195.5 | 224.2 | 200.7 | 176.2 | 199.5 | 260.8 |
| | $R^2$ | 0.9908 | 0.9955 | 0.9956 | 0.9948 | 0.9926 | 0.9907 | 0.9996 | 0.9905 |
| Inter-particle diffusion * | $^1 k_p / \text{mg g}^{-1} \min^{-\frac{1}{2}}$ | 26.27 | 23.26 | 38.36 | 43.86 | 43.51 | 21.14 | 33.31 | 56.90 |
| | $^2 k_p / \text{mg g}^{-1} \min^{-\frac{1}{2}}$ | 3.61 | — | — | 4.06 | 3.69 | — | — | 7.41 |
| | $^1 R^2$ | 0.9647 | 0.9799 | 0.9551 | 0.9851 | 0.9752 | 0.9020 | 0.9388 | 0.9554 |
| | $^2 R^2$ | 0.9994 | — | — | 0.9757 | 0.9760 | — | — | 0.8265 |

* Note: $^1 k_p$ and $^2 k_p$ and $^1 R^2$ and $^2 R^2$ are the interparticle diffusion model rate constants and correlation coefficients of two sections of diffusion [1] before and [2] after breakpoint.

Furthermore, it seems that the kinetic constant $k_2$ increases in the co-action of BAC with GAC than in the action of these GAC alone. For example, rate constant $k_2 = 1.0 \times 10^{-3}$ of *flufa* adsorption on GAC alone increased at $k_2 = 3.1 \times 10^{-3}$ in co-action with BAC. The increase of $k_2$ corresponds to the rapid ion pair formation using BAC (according to Scheme 1 and Figure 4).

On the other hand, adsorption is a multistage process. It can also involve the transfer of pollutant molecules from the aqueous solution to the surface of the adsorbent and subsequently penetrate into pores [35]. The PFO and PSO kinetics models are not able to describe this complex diffusion mechanism [35]. Therefore, the interparticle diffusion model (the Weber–Morris model) was applied to the evaluation of the diffusion processes [32,33].

Table 4 presents the evaluation of the interparticle diffusion model, providing the correlation coefficients $R^2$ higher than 0.9. However, the curves of the interparticle diffusion model of virgin GAC are not linear over the whole-time range; see Figure S7A,B in the Supplementary Materials. There are two sections of diffusion before and after the so-called breakpoint time (interception of the two curves).

**Table 4.** Column adsorption characteristics.

| Modification-Sorption Cycle No. | Breakthrough Time/h | Flow Rate/cm$^3$ min$^{-1}$ | Bed-Depth/cm | $q_{column}$/mg g$^{-1}$ |
|---|---|---|---|---|
| 1 | 4.96 | 8.4 | 6.5 | 54.2 |
| 2 | 4.46 | 8.4 | 6.5 | 36.2 |
| 3 | 4.46 | 8.4 | 6.5 | 46.6 |
| 4 | 2.48 | 8.4 | 6.5 | 23.0 |
| 5 | 1.49 | 8.4 | 6.5 | 10.4 |

It can possibly indicate that several processes influenced the adsorption [35]. Probably, intraparticle diffusion is not the sole rate-limiting mechanism. Other kinetic models may simultaneously control the rate of adsorption [36]. The mentioned breakpoint time shows the potential difference in adsorption diffusion mechanisms [36].

In addition, we observed different mechanisms of initial adsorption within the adsorption of *flufa* on virgin GAC alone, and after the breakpoint, it rather occurred by interparticle diffusion. On the other hand, in the case of co-action of GAC with BAC, the opposite effect was observed; see Figure S7A,B in the Supplementary Materials. This fact may indicate that the co-action of BAC and GAC enables better utilization of the available specific surface area of GAC. Nevertheless, it is difficult to exactly evaluate the complex adsorption mechanisms.

### 3.4. Comparison of Enhanced Adsorption Using BAC in Column and Batch Experiments

As we observed earlier (see Figure 5), the tested sample of spent-GAC allows lower removal efficiency and adsorption capacities for both *flufa* and *dcf* than virgin GAC. Therefore, further use of such depleted GAC appears to be unattractive. Finally, the batch and column

arrangements were compared with an emphasis on a potential practical application of BAC within enhancing the spent-GAC adsorption capacity. These laboratory experiments were tested within the *flufa* adsorption at an initial concentration $c_0$ = 2 mmol L$^{-1}$ (562 mg L$^{-1}$), simulating the real wastewater produced during drug manufacturing [42].

We observed that spent-GAC is almost ineffective ($c_{effluent}/c_{influent}$ > 0.9) for *flufa* separation from concentrated aqueous streams in fixed-bed column arrangements; see 1st-3rd points in Figure 6. Compared to the batch experiments (e.g., in Figure 5), it can be caused by less intense and shorter contact time of *flufa* and spent-GAC.

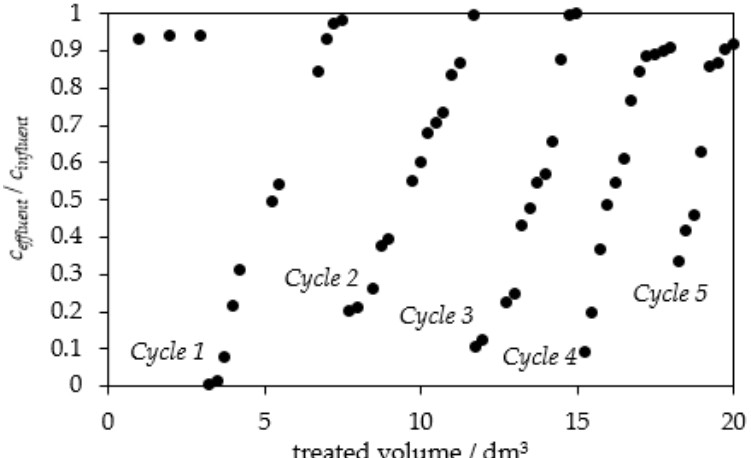

**Figure 6.** Removal of *flufa* from aq. solution ($c_0$ = 562 mg L$^{-1}$) on GAC (25 g on the fixed-bed column) with activation of saturated GAC using 64 mg 50% aq. BAC per 1 g of spent-GAC.

Effective repeated utilization of the adsorbent material is crucial according to the circular economy principles [36–38]. The purpose of spent sorbent regeneration is to remove the loaded adsorbate (pollutants) from the column in a small volume of eluates [38]. However, the mentioned regeneration should produce concentrates of pollutants suitable for smooth degradation without loss of the adsorption capacity of the adsorbent and make it reusable in several adsorption and desorption cycles. Furthermore, regeneration should also ensure that concentrated streams do not produce any disposal problems [38].

On the basis of the mentioned requirements on the effective utilization of sorbent, the enhancement of the adsorption capacity of GAC without elution of *flufa* from the fixed-bed column was scrutinized. Therefore, the above-verified and effective BAC was applied as a spent-GAC activator.

After the first application of BAC onto the spent-GAC in the column and its percolation, the subsequent *flufa* adsorption proved to rapidly increase the removal efficiency ($c_{effluent}/c_{influent}$ < 0.05); see Figure 6. In this first activation-sorption cycle, the breakthrough time of *flufa* was 4.96 h, and the adsorption column capacity was 54.2 mg g$^{-1}$, dealing with a flow rate of 8.4 cm$^3$ min$^{-1}$; see Table 4. The next two activation-sorption cycles provided similar results. In other cycles, the breakthrough time and column adsorption capacity decreased.

Nevertheless, after 4th activation of exhausted GAC using BAC, a higher than 80% increase in *flufa* separation was observed. The total adsorption capacity of GAC for *flufa* after five activations-sorption cycles in the fixed-bed column was $\Sigma q_{column}$ = 170.4 mg g$^{-1}$.

It can be assumed that potential other cycles of activation of GAC in the column could be performed using higher doses of BAC within activation steps. The price of applied BAC [29] is still lower than the pyrolytic regeneration of GAC [55], and it is obvious that the prolonging of GAC lifetime using BAC is able to postpone expensive GAC regeneration.

With benefit, the *flufa* were not detected in the eluates after percolation of 50% aqueous solution of BAC within the activation step. It indicates that on the surface of GAC, the ion exchange between *flufa* and BAC proceeds according to the reaction in Scheme 1. This mechanism probably enables subsequent entrapment of *flufa* on GAC activated with BAC.

It provides the increasing of GAC adsorption capacity in the column without elution of *flufa* and ongoing degradation of drug concentrates after activation cycles.

Compared to the column experiments, the batch arrangements of *flufa* repeated adsorption on spent-GAC in co-action with BAC were investigated. As Figure 7 presented, spent-GAC (4 g L$^{-1}$) provides 33% removal efficiency of *flufa* from concentrated model wastewater ($c_0$ = 562 mg L$^{-1}$) in the first cycle (in Figure 7 cycle 0) of *flufa* adsorption. Then, the removal efficiency of *flufa* using reused spent-GAC decreased to ca. 20%, and after seven subsequent cycles of repeated adsorption, the removal efficiency of *flufa* was lower than 20%.

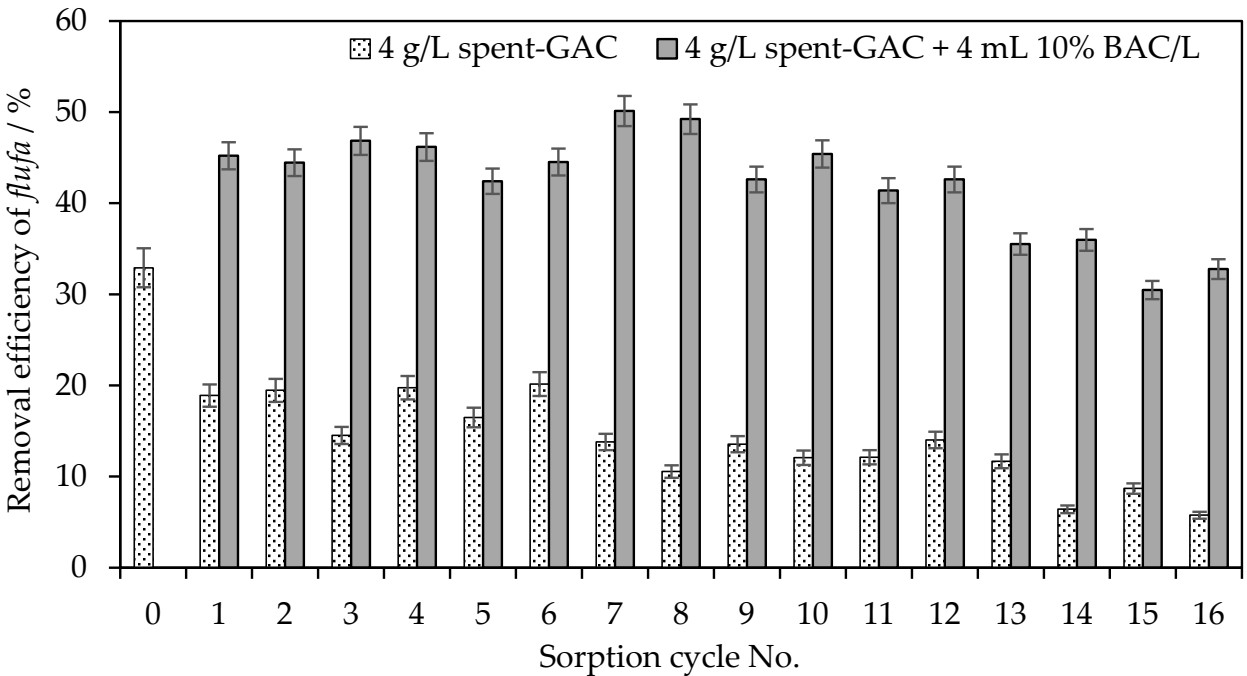

**Figure 7.** A batch adsorption of *flufa* on spent-GAC and spent-GAC in co-action with BAC (30 min of vigorous agitation was used in each sorption cycle).

However, the application of BAC provides an increase in *flufa* removal from 20% to over 45% (sorption cycle 1); see Figure 7. The removal efficiencies of *flufa* higher than 45% were observed in the subsequent 12 repeated cycles. Then, the efficiency of *flufa* separation slightly decreased. However, the entrapment of *flufa* on spent-GAC with the addition of BAC was still almost about 30% higher than the adsorption on spent-GAC alone, even in the 16th cycle of *flufa* adsorption; see Figure 7.

The co-action of BAC with spent-GAC increased the adsorption capacity of spent-GAC by 560.4 mg g$^{-1}$ (q$_{spent-GAC+BAC}$ = 760.2 mg g$^{-1}$ within removal efficiency of *flufa* > 40%, cycles 1–12; and q$_{spent-GAC}$ = 199.8 mg g$^{-1}$; within removal efficiency of *flufa* > 20%, cycles 0–6).

The batch arrangements provided apparently better results in enhancing the adsorption capacity of spent-GAC compared to the fixed-bed column experiments. However, a direct comparison of batch and column arrangements is not possible due to the different performances of experiments. The main difference between column and batch experiments can also be the proposed mechanism of modification of GAC using BAC. As we described above, the in situ activation of GAC with BAC is based on rapid ion pair formation and subsequent adsorption of respective less polar ion pairs on a rather non-polar surface of GAC.

## 4. Conclusions

In this study, we investigated the removal of halogenated pharmaceuticals represented with *flufa* and *dcf* with respect to the enhancement of adsorption capacities of GAC using available and inexpensive BAC.

Though spent-GAC is not able to remove pollutants from water effectively, a co-action of spent-GAC and BAC increased the removal efficiencies of drugs up to 25% compared to the use of spent-GAC alone.

Based on the kinetic model evaluation, the adsorption results were the most consistent with the pseudo-secondary kinetic model (PSO). The adsorption capacity according to the PSO of spent-GAC in co-action with BAC ($q_{flufa}$ = 195.5 mg g$^{-1}$ and $q_{dcf}$ = 199.5 mg g$^{-1}$) reached the adsorption capacity of virgin GAC ($q_{flufa}$ = 203.9 mg g$^{-1}$ and $q_{dcf}$ = 200.7 mg g$^{-1}$). In addition, the rate constant $k_2$ increases in the co-action of GAC with BAC. We evaluated that the rate constant $k_2 = 1.0 \times 10^{-3}$ of *flufa* adsorption determined by the action of GAC alone increased at $k_2 = 3.1 \times 10^{-3}$ in co-action of BAC and GAC.

Finally, a comparison of column and batch experiments with emphasis on the potential practical application of BAC showed that the adsorption capacity of spent-GAC within several cycles of repeated adsorption of *flufa* can be enhanced using tested BAC. In both column and batch experiments, the adsorption capacity of spent granulated AC for *flufa* increased using BAC by 170.4 mg g$^{-1}$ and 560.4 mg g$^{-1}$, respectively.

The increasing adsorption capacity is probably caused by the formation of low polar and poorly soluble ion pairs (BAC-drug) and subsequent adsorption of these low polar ion pairs on the surface of GAC. The lower polarity of ion pairs corresponds well with the octan-1-ol/water coefficients $P_{OW}$ for BAC-*dcf* and BAC-*flufa* (3.26 and 5.13, respectively). The $^1$H NMR confirmed the formation of the above-mentioned ion pairs.

In conclusion, the co-action of BAC with spent-GAC can enhance the treatment effect of high-concentrated wastewater streams containing acidic polar pharmaceuticals and provide innovative ideas for the extension of GAC lifetime within a batch as well as column adsorption processes.

**Supplementary Materials:** The following supporting information can be downloaded at: https://www.mdpi.com/article/10.3390/w15183178/s1, Figure S1. A CV-voltammogram of *flufa* (A) and CV-voltammogram of the sample after leaching of spent-GAC in water (B) [56].; Figure S2. A partition coefficients octan-1-ol/water of sodium salts of *flufa* and *dcf* (drug-Na) and corresponding ion pairs (BAC-drug); Figure S3. $^1$H NMR spectra (A–C) of isolated ion pair BAC-*dcf*; Figure S4. $^1$H NMR spectra (A–C) of isolated ion pair BAC-*flufa*$^-$; Figure S5A. Effect of the modification mechanism on separation of *flufa* ($c_0$ = 100 mg L$^{-1}$) after 20 h of action; Figure S5B. Graphical presentation of impregnation/modification methods of GAC using BAC within *flufa* adsorption; Figure S6. The fitting of experimental data with PFO and PSO models for GAC (A), GAC + BAC (B), spent-GAC (C), and spent-GAC + BAC (D) within *flufa* adsorption; Figure S7. The linear plots of the interparticle diffusion model for adsorption of *flufa* on (A) GAC and (B) GAC+BAC; Figure S8a. The ED XRF spectra of light elements spectral region (BK3: virgin GAC (Hydraffin CC8x30); BK4: spent-GAC; BK5: virgin GAC saturated with DCF; BK6: virgin GAC saturated with ion pair BAC-DCF); Figure S8b. The ED XRF spectra of heavy elements spectral region (BK3: virgin GAC (Hydraffin CC8x30); BK4: spent-GAC; BK5: virgin GAC saturated with DCF; BK6: virgin GAC saturated with ion pair BAC-DCF). Scheme S1. The experimental scope of adsorption of tested drugs on GACs alone or in co-action with BAC.

**Author Contributions:** Conceptualization, B.K. and T.W.; methodology, T.W.; software, B.K.; validation, B.K. and M.P.; formal analysis, B.K. and M.P.; investigation, B.K., M.P. and T.W.; resources, B.K.; data curation, B.K. and M.P.; writing—original draft preparation, B.K.; writing—review and editing, T.W.; visualization, B.K.; supervision, T.W. All authors have read and agreed to the published version of the manuscript.

**Funding:** This research received no external funding.

**Data Availability Statement:** Not applicable.

**Acknowledgments:** The authors acknowledge the financial support for excellent technological teams from the Faculty of Chemical Technology, University of Pardubice. The authors also acknowledge Pavel Matějíček for performing several experiments (in Figures 6 and 7) as a part of his thesis.

**Conflicts of Interest:** The authors declare no conflict of interest.

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
