# Peer review of "Sorption of Halogenated Anti-Inflammatory Pharmaceuticals from Polluted Aqueous Streams on Activated Carbon: Lifetime Extension of Sorbent Caused by Benzalkonium Chloride Action"

_water, doi:10.3390/w15183178_

Round 1
Reviewer 1 Report
- In the Introduction, the authors state that modification is one option to decrease the economic aspect of regeneration. The phrase “ economic aspect” needs to be revised with another proper one, such as ‘reducing cost’, etc. It is recommended to get professional proofreading to increase the readability of this manuscript;
This sentence in Introduction is also confusing “The modification of AC can prolong the lifetime of this effective, but also expensive adsorbent.”
2. Characterization of the spent-AC adsorbent is essential to define the used-AC itself. An XRD, FTIR, and elemental analysis by XRF of EDX will inform the used-AC in this research;
- In Result and Discussion, authors informed about removal efficiency (%), and adsorption rate. But that discussion will be weak unless some supporting primary data are provided in Discussion—the primary data to prove the presence of the adsorbates before and after adsorption.
- The presence of adsorbates within the AC after adsorption needs to be analyzed. Investigating the change of the available functional groups before and after adsorption is essential. Authors shall do FTIR analysis for this objective.
some sentences need to be revised to increase readability
Author Response
Dear Reviewer,
we thank you for such an opinion and for valuable comments. We have done our best to reflect all your queries and recommendations, incorporating your comments in the new version of article. The research design, methods and results parts were improved, please see chapters 2-3. All these changes/improvements in new version of article are marked yellow.
Answer to the comment 1:
Your comments for some phrases and other issues were revised in an article. Moreover, the article was revised and corrected.
Answer to the comments 2. and 3:
GAC samples before and after adsorption of DCF were characterized using XRF and FTIR. XRF measurements approved significantly higher content of chlorine in GAC samples saturated with Diclofenac (please, see Figures S8a and S8b in Supplementary Materials).
Unfortunately, FTIR spectra measured by the DRIFT technique provided no detectable signals of adsorbed Diclofenac.
However, more extensive study focused on this topic could be the subject of our future work.
Reviewer 2 Report
Dear Authors,
The manuscript entitled “Sorption of Halogenated Anti-inflammatory Pharmaceuticals from Polluted Aqueous Streams on Activated Carbon: Lifetime Extension of Sorbent Caused by Benzalkonium Chloride Action” presents the studies that fit well with the journal scope. The manuscript needs amendments to be published.
Chapter 2.1 contains not only a description of chemicals and reagents but also the characteristics of adsorbents. It should be mentioned in the title of the chapter, e.g. Chemicals and adsorbents. But better divide it into two separate chapters. What is the difference between chemicals and reagents in terms of the chapter’s title?
I think the reference no 1 is improperly used. You wrote about the environmental problems of occurring this type of contaminant but you used as the reference the article about adsorption. It should be modified. Lines 30 and 36.
Comparing figures 3 and 4 it’s clear that BAC impregnation gives not the best results. But I couldn’t see clear this conclusion in chapter 3.2.
The order of information is not clear. First, the authors present the results of removal efficiency in the time (fig 4) and next, inform about the preliminary tests and several options (fig. S4-A). After reading I couldn’t get to know what method was used for the tests the results are presented in Fig. 4.
Lines 333-337 Nowadays, the assumptions of the physical and chemical nature of PFO and PSO models have been challenged.
There are several testing paths in the research scope, in terms of materials, contaminants and procedures. After reading the methodology chapter it’s not clear and for me, it was hard to build a whole experiment design in my mind in a short time. So, I recommend adding a scheme of experiments with the main conditions.
I’m almost sure, the Authors modelled all data points from kinetic experiments, even from the equilibrium. As evidenced, only the data from not complete uptake should be used for the analysis. Ref: doi.org/10.1016/j.cej.2016.04.079; 10.1021/acs.iecr.7b04724; doi.org/10.1016/j.watres.2017.04.014. If the Authors modelled total data sets, please do it again. Especially, the PFO model shown in Figure S5 looks optimised incorrectly. The Authors can also consider this assumption for kinetics data modelling doi.org/10.3390/w15061231
The results of the intraparticle diffusion model (Table 3) contain only one rate (kp) when Figure S6 presents two sections of diffusion. So, there should be two separate rates.
Best regards,
Author Response
Dear Reviewer,
we thank you very much for your opinion and for valuable comments. We have done our best to reflect all your queries and recommendations, incorporating your comments in the new version of article. All changes in new version of article are marked yellow.
Answers of authors to the comments:
1) There is no difference between chemicals and reagents in terms of 2.1 chapter´s title. Now, chapter 2.1 is divided into two separate chapters – 2.1 Chemicals (describing used chemicals) and 2.2 Adsorbents (describing used adsorbents and their specifications). Please, see new version of article with divided mentioned chapters.
2) Yes, you are right. Reference no 1 was improperly used. It was probably caused by wrong inclusion of citations in the final draft of article. Thank you for this point, now it should be clear. See Introduction part and also References.
3+4) Graph 3 (chapter 3.1) presented the adsorption of drugs on virgin PAC and GAC and exhausted GAC (spent-GAC) alone. Afterwards, Graph 4 in chapter 3.2 depicted the rate of drugs removal using BAC alone (not the impregnation of AC using BAC). As we published earlier [41,50], the separation of acid drugs or dyes using quaternary ammonium compounds (surfactants or ionic liquids) is very effective. The ion exchange of R4NX with acid drugs (according to Equation 9) provided less polar ion pairs (Contaminant-COONR4) that can be separated from aqueous solutions by sedimentation and/or filtration. The rate of described separation of drugs using R4NX (BAC) alone presented mentioned graph 4.
Contaminant-COONa + R4NX → Contaminant-COONR4 + NaX |
(9) |
The results of separation of drugs using BAC are described in introduction of chapter 3.2. The action of BAC applied for removal of tested drugs provided only limited removal efficiencies (ca. 50%); see Fig. 4. On the other hand, the co-action (in situ impregnation) of spent-GAC (exhausted granulated AC) with BAC enable almost comparable removal efficiencies such as virgin GAC alone; see Figure 5 in chapter 3.2. Moreover, the best results of drugs separation provided the co-action of virgin GAC and BAC.
Following up on the previous comment, the order of information is as follows:
- Adsorption of drugs on adsorbents alone (see chapter 3.1 and Figures 2 and 3)
- Separation of drugs using BAC alone (see introduction of chapter 3.2 and Figure 4 and NMR spectra in Supplementary Materials)
- The comparison of several impregnation methods of GAC/spent-GAC using BAC = co-action of ACs and BAC (see chapter 3.2 and Fig S4A-B)
- And finally, the adsorption kinetics study within adsorption of drugs on GAC alone, spent-GAC alone, co-action of GAC with BAC, and co-action of spent-GAC with BAC (see new chapter 3 Adsorption Kinetics Study, and Figure 5 and Table 3; the research scope of these experiments is now presented in new Scheme S1 in Supplementary Materials).
We improved the title of Figure 4 and description of results presented in mentioned Figure 4 For the better clarity of the discussed results. We improved the description of removal efficiencies of drugs within several impregnation methods of GAC using BAC, see chapter 3.2 and Fig. S4-A. We also divided the different methods of GAC impregnation and adsorption kinetics study into 2 chapter. Please, check out the new version of article, especially chapter 3.2 and new chapter 3.3.
5) Thank you for reporting of new trends and clarification of the nature of PFO and PSO models. This new knowledge was now included in an article with relevant (recommended) reference.
6) We tried to improve the clarity of research scope and we added the scheme of experiments with the main conditions. Please, see chapters 3.2 and 3.3 and new Scheme S1 in Supplementary Materials.
7) Thank you very much, the recommended references were very helpful for kinetics data modelling improvement. The modelling of data according to PFO and PSO models was revised, and data were corrected according to the recommended citations. Please, see Table 3 and Figure S6. In addition, we also involved some assumptions for kinetics data modelling described in recommended citation doi.org/10.3390/w15061231.
8) In the sense of previous comment (7), we revised the data evaluation of the interparticle diffusion model. The two rates kp of intraparticle diffusion model (before and after breakpoint) were revised and added in Table 3.
Best regards,
Barbora, Miloslav and Tomáš

Reviewer 3 Report
The proposed manuscript, titled "Sorption of Halogenated Anti-inflammatory Pharmaceuticals from Polluted Aqueous Streams on Activated Carbon: Lifetime Extension of Sorbent Caused by Benzalkonium Chloride Action," aims to explore the adsorption of halogenated anti-inflammatory drugs using activated carbon modified with benzalkonium chloride. The manuscript is skillfully written, providing a strong introduction that addresses the imperative of developing innovative and cost-effective approaches for treating wastewater and removing toxic pollutants.
The study is well-designed, encompassing a significant number of experimental conditions and investigations. The results are compellingly supported with ample experimental data. This innovative approach presents a promising alternative to reduce or eliminate pollutants like NSAIDs from wastewater, rendering it both environmentally friendly and cost-effective. I recommend accepting the manuscript for publication in Water Journal in the present form.
Author Response
Dear Reviewer,
we thank you for such an opinion, and we are delighted with your decision.
Best regards,
Barbora, Miloslav and Tomáš

Round 2
Reviewer 1 Report
The authors revised the article by addressing all on the reviewer's comments
Reviewer 2 Report
Thank you, the authors have corrected the manuscript sufficiently